# The Spatio-Temporal Patterns and Driving Forces of Land Use in the Context of Urbanization in China: Evidence from Nanchang City

**DOI:** 10.3390/ijerph20032330

**Published:** 2023-01-28

**Authors:** Yuxi Liu, Cheng Huang, Lvshui Zhang

**Affiliations:** 1School of Forestry, Jiangxi Agricultural University, Nanchang 330045, China; 2Shanghai Key Lab for Urban Ecological Processes and Eco-Restoration, School of Ecological and Environmental Sciences, East China Normal University, Shanghai 200241, China

**Keywords:** land use change, drivers, urban expansion, landscape

## Abstract

Land use change has been one of the common problems in the context of urbanization in China. Social economy and land use interact with each other, and it is especially important for human society to adhere to sustainable development, and to deal with the contradictory relationship between the social–economic needs and land use change. The objectives of this study are: (1) Obtain time-series land-use classification data and its spatial distribution in Nanchang City; (2) Identify the characteristics and driving force of spatial–temporal land use changes in Nanchang City from 2000 to 2020; (3) Discuss the relationship between the urban expansion and social economy in Nanchang City. The results show that the spatial distribution of land use in Nanchang City has changed significantly from 2000 to 2020, and the largest area of land-use type in Nanchang City has been cropland. The cropland has continuously declined, and the urban area has increased significantly. A lot of cropland has been transformed into urban areas, and land use degree in Nanchang City has significantly increased. The spatial pattern of land use has greatly changed, and the city spatial pattern has become more aggregated, while the spatial distribution of cropland, forest and grassland has become more fragmented. Moreover, there has been an obvious correlation between social-economic development and the level of land use, and GDP has been the main driver of land use change. The central urban area of Nanchang city has been the main hotspot of land use change.

## 1. Introduction

Land use is a complex system containing natural elements such as geology, hydrology, soil, vegetation and human activities [1]. It is closely related to social–economic development [2]. Land use change is a reflection of human activities and a key factor influencing climate change [3,4]. Land-use change mechanisms, monitoring and modeling have become the focus of the scientific community [5,6]. Land use surveys, land resource assessment and utilization planning are the application fields of traditional land-use research [7]. Socio-economic, cultural and spatial land management make land use show significant regional and periodic differences [8,9].

Since China officially became a member of the World Trade Organization (WTO) in 2001, China’s economy and urbanization processes have entered a stage of rapid development. Urbanization is considered to have a significant impact on land use patterns and the environment [10]. So far, more than 54% of the world’s total population (about 8 billion) lives in urban areas and this figure is expected to exceed 67% in 2050. Urbanization is placing an increasing demand on urban land, yet the demand for land resources for urbanization cannot be met indefinitely as land resources are scarce resources [11]. How to use land resources efficiently and intensively has become one of the issues of sustainable development in human society [12]. Clarifying the mechanisms of interaction between land use change and social economy is important for the formulation of urban sustainable development strategies and the sustainable use of land resources.

Urbanization is one of the most important factors of LULC change, and it also brings up a series of ecological problems [13], such as the heat island effect [14], environmental pollution [15], habitat destruction [16] and loss of cropland [17]. Urban expansion is one of the essential parts of land use research. Since the 1820s, researchers have focused mainly on urban morphology and spatial structure in land use [18]. Suggested theories for urban expansion include the concentric model, sector theory and multi-core model [19]. With the continuous development of remote sensing and geographic information science (GIS) technology, the research on urban expansion based on the perspective of land use has developed into quantitative analysis, dynamic monitoring, spatiotemporal characteristic analysis and driving mechanism research of urban expansion [18].

Due to the difficulty of obtaining time-series spatial data, the early land use research mainly discussed the area change of land use from a statistical perspective, so the spatial pattern change of land use was not fully studied. The development of remote sensing technology and computer technology has provided rich monitoring methods and efficient data-processing methods for land use research. Additionally, the application of remote sensing technology not only provides rich and diverse spatial monitoring data but also greatly saves human and financial costs. Liu [20] showed 13 cities in China that urban expansion is influenced by a combination of population, monetary and land policies. Danilo [21] used an object-oriented classification to analyze aerial photographs and obtain land use situations in the Langhe region of Italy from 1954 to 2000. He also further analyzed changes in the spatial–temporal pattern of land use. Although land use data have good spatial resolution and effectively promote the study of urban space, it is mainly based on Landsat data interpretation, which relies on traditional methods that are time-consuming and labor-intensive. With the progress of Big Data technology and the abundance of remote sensing data, the rapid and low-cost land use data acquisition of time series is possible, and the joint research of remote sensing data and socio-economic time series is gradually enriched [22].

To solve the problem of discontinuous land-use data and obtain urban spatial distribution data quickly, researchers have turned their attention to night-time lighting data from meteorological satellites [23,24,25]. However, the inherent disadvantages of night-time light data, such as low resolution (1 km), determining that the data are suitable for the extraction of urban boundary information and lack more detail, make the data unsuitable for comprehensive land-use analysis and monitoring.

Advances in remote sensing and artificial intelligence technologies have dramatically changed the way we look at the Earth [12]. The Remote Sensing Big Data Platform (RSBDP) provides a powerful and data-rich platform for remote sensing analysis for researchers and provides remote sensing analysis functions, such as GEE [26]. Currently, RSBDP has been widely used in a variety of fields such as land cover mapping, and Hanberry [27] noted that RSBDP and machine learning methods performed well in wildfire identification and classification. Some researchers found that Landsat images and GEE were suitable for mapping flood areas, with an overall accuracy of over 90% [28]. The researchers found that by using machine learning methods and Landsat data, artificial targets in heterogeneous environments could be accurately detected [29]. The researchers found that GEE and random forest classifier can effectively extract vegetation distribution information [30]. RSBDP-based land-use classification methods and statistical characterization are hot issues in land use research [31].

In this study, we propose to use spatiotemporal Big Data and time-series analysis methods to explore the driving force of land-use pattern change in Nanchang from the perspective of spatiotemporal change, so as to provide decision-making support for sustainable use of land resources and ecological environmental protection. A large number of existing studies of land use cannot quantify the correlation between land use change and social economy using statistical methods due to the lack of continuous time-series data on land use. The innovations of this study are to quantify the correlation between land use change and social economy using continuous time-series land use data and statistical methods. In addition, the correlation among comprehensive land use index, per capita GDP, population and fixed asset investment was analyzed in this study.

## 2. Materials and Methods

### 2.1. Research Areas

Nanchang is the capital city of Jiangxi Province and an important city in the urban agglomeration in the middle reaches of the Yangtze River in China. It is also the central city of the Poyang Lake Ecological Economic Zone. Located in the north-central of Jiangxi Province (Figure 1), it covers an area of 7194.98 km^2^. As of 2020, Nanchang has a resident population of 6,437,500 and a GDP of 574,551 million CNY. In addition, Nanchang is one of the first low-carbon pilot cities in China, a national innovative city and an international garden city. In recent years, the economy of central China was growing rapidly at a rate of about 8%, leading to increased urbanization and rapid land use changes, which have had an impact on urban spatial form and the ecological environment. Therefore, this study analyzes the relationship between land use patterns and social–economic development in Nanchang City over the past 20 years and provides decision support for urban green development and sustainable development.

### 2.2. Materials

This study mainly uses statistical data and spatial data. Among them, the statistical data were obtained from the Nanchang Statistical Yearbook (2001–2021), including population, GDP and fixed investment. Spatial data include land use and administrative division data. Among them, land use data were obtained by using remote sensing Big Data platform, selecting Landsat impact data and using a supervised classification method and machine learning algorithm to classify remote sensing images. Benefiting from the application of the RSBDP and machine learning algorithm, this study can quickly obtain continuous time-series land-use data and carry out innovative research. The administrative division data were obtained from the China Geographic Information Bureau and were produced in 2015. This study classifies land use in Nanchang into seven types: cropland, forest, shrub, grassland, water, barren and urban.

### 2.3. Methods

The research framework is shown in Figure 2. Previous studies on urban land use and drivers were limited by the lack of spatial data on land use and mainly used discontinuous land use data every 5 years for analysis. Therefore, previous studies could not fully utilize the statistical data of continuous time series to mine the relationship between land use and the social economy, nor could they construct statistical models of social economy and land use. They could only analyze the social-economic impact on land use in the datum year and the target year through LMDI or other methods. In recent years, thanks to the maturity of remote sensing Big Data and remote sensing cloud-computing platforms, the acquisition and analysis of spatial–temporal data based on the remote sensing Big Data platform becomes a reality, and the spatial–temporal data of continuous time series can be quickly acquired and applied to practice research. As a result, integrated research of spatial–temporal change and social economy based on time series can be carried out smoothly.

#### 2.3.1. Land Use Classification

This study uses remote sensing Big Data analysis platform to obtain time series Landsat satellite images, and then supervised and classified them by combining random forest and transfer learning methods to obtain the land use data of Nanchang City from 2000 to 2020. The land use classification process is shown in Figure 2. Firstly, we used the 4 periods of land use data (2000, 2005, 2010, 2015) released by the Institute of Geographical Sciences and Resources, Chinese Academy of Sciences (https://www.resdc.cn/Default.aspx, accessed on 27 March 2021) to randomly collect unchanged stable regions form the 4 periods of land use data as training samples for supervised classification, and made them into sample data labels. Then, we chose seven types of land use samples: cropland, forest, grassland, water, barren and urban. Secondly, we screened the Landsat dataset of 2000–2020 using RSBDP, calculated NDVI, MNDWI and NDBI and used the results as the three bands. We then exported them as a new dataset. After that, the sample data labels were imported into the RSBDP platform and were extracted for the sample dataset to train the Random Forest classifier. We used the trained Random Forest classifier to supervise and classify the Landsat dataset from 2000 to 2020. Finally, we implemented spatial–temporal filtering and accuracy assessment on the classification results, and if the classification results did not meet the accuracy requirements, the sample labels were modified. The classification operation was repeated until the classification accuracy was more than 80%, and then the classification was finished. In this study, the classification method was the same as the existing literature, and more detail can be found in the literature [32].

#### 2.3.2. Land Use Comprehensive Index

The quantitative study of land use degree is based on its limit, that is, when land resource utilization reaches the limit, human beings will not able to continue developing land resources. The lower limit of land resource use is the starting point for human exploitation of land resources [33]. The magnitude of the composite index reflects the level of land use. In order to quantitatively evaluate the degree of land use, the existing research classifies the ideal state of land use into four levels, and gives corresponding weights to the four levels of land use, so as to achieve a quantitative evaluation of the land use degree. This study uses weight assignment of land use degree from the existing studies to calculate the comprehensive index of land use in Nanchang (Table 1).

The comprehensive index of land use is a Weaver index, and its calculation model is shown below [33]:(1)$$L_{i}=100\times \sum_{j=1}^{n}A_{j}\times P_{j}$$
where *L_i_* is the comprehensive index of land use with values in the range (100, 400); *A_j_* is the graded index of high land use at level *j* and *P_j_* is the percentage of the graded area of land use degree at level *j*.

#### 2.3.3. Relative Change Rate of Land Use

In order to quantitatively analyze the different characteristics of land use change in each district and county of Nanchang City, we introduced the index of land-use relative change rate. The relative change rate of land-use type change is built on the basis of the change rate index. It is the ratio of the change rate of land use type in target areas to the change rate of land use type change in the study area, which is a hot spot area for analyzing differences in specific land use type change and specific land use type change within the study area [34]. The calculation model of the relative change rate of land use is as follows:(2)$$K=\frac{\left|K_{t+1}-K_{t}\right|\times C_{t}}{K_{t}\times |C_{t+1}-C_{t}|}$$
where *K_t_*_+1_, *K_t_* are the area of a particular land use type in the target areas at moment *t*+1 and t, respectively; *C_t_*_+1_, *C_t_* are the area of the land use type in the study area at moment *t*+1 and *t*, respectively. The significance of the absolute values is to avoid confusion caused by the direction of land use change and to facilitate comparison between target areas. The significance of the relative change rate is to reveal differences in the land use change of study areas [35].

#### 2.3.4. Landscape Index Analysis

In order to deeply understand the spatial variation characteristics of land use, we selected four landscape-scale indicators, Shannon’s Evenness Index (SHEI), Patch Density (PD), Contagion Index (CONTAG) and Number of Patches (PD) for the study. We also chose Total Landscape Area (CA), Largest Patch Index (LPI), Patch Cohesion Index (COHESION) and Splitting Index (SPLIT) to quantitatively evaluate the spatial pattern of land use in Nanchang City. The above indicators were calculated using Fragstats version 4.2 (Gen. Tech. Rep. PNW-GTR-351. Portland), a landscape pattern analysis software.

## 3. Results and Discussion

### 3.1. Land Use Change

The results showed the overall accuracy of land use was 88.36%, and the Kappa index was 42.49%. From 2000 to 2020, the spatial distribution of land use in Nanchang City has changed significantly (Figure 3). Among them, urban expansion has been significant, especially in the central city of Nanchang and Honggutan. Cropland has always been the largest land use type in Nanchang.

In the last 21 years, the area of cropland in Nanchang has decreased by 2.94%, from 66.74% to 63.80% (Figure 4a). The loss of cropland area has mainly devolved into urban (51.49%), forest (28.86%) and water (19.15%) areas (Figure 4b). In order to protect the ecological environment, the Chinese government has implemented the project of “returning farmland to lakes” in the middle and lower reaches of the Yangtze River. Therefore, part of the cultivated land around Poyang Lake has become wetland. Urban expansion was the main factor leading to the loss of cropland, which was consistent with the conclusions of existing studies that the urbanization process is the main factor leading to the massive loss of cropland. From 2000 to 2020, the urban area has increased from 379.15 km^2^ to 722.66 km^2^, an increase of 90.60% (Figure 4a; Table 2). Of this increase, 87.25% of the urban area from cropland to urban, 8.09% from water areas and 3.10% from forest (Figure 4b). It is worth noting that the forest area of Nanchang City has increased from 802.25 km^2^ in 2000 to 842.67 km^2^ in 2020, an increase of 5.04%. In addition, Grassland, Water and Barren areas have decreased by 4.58 km^2^ (48.82%), 160.52 km^2^ (12.75%) and 1.22 km^2^ (10.40%), respectively (Table 2). Urban expansion has been the main reason for the decrease in grassland area, with 2.81 km^2^ of grassland converted to urban. On the other hand, Barren areas have decreased by 1.22 km^2^ from 2000 to 2020, of which 0.91 km^2^ has been converted to Grassland. Most of it has transformed into grassland in urban parks. A total of 2.90 km^2^ of Barren areas have been converted into urban. Therefore, urban expansion has made dramatic changes in Barren areas.

Figure 5 shows the variation trend of urban, cropland and forest areas in the nine administrative districts of Nanchang. The urban expansion in Honggutan was the most significant, increasing from 13.48 km^2^ to 63.19 km^2^, an increase of 368.75% (Figure 5a), followed by Qingshanhu and Xinjian County, with an increased urban area of 92.76% and 74.02%, respectively. Honggutan has been under construction since 2000 and has experienced a high growth period from 2013 to 2016, with an average annual expansion of 5.18 km^2^. The district with the smallest urban area expansion was Donghu (15.06%), followed by Qingyunpu (44.39%), Jinxian (51.68%) and Xihu (54.68%). Donghu, Qingyunpu and Xihu were the central urban areas of Nanchang City with a high degree of land development, and therefore the urban area has increased less in the past 21 years. Jinxian was a suburban administrative district of Nanchang City and was located at the periphery of the central urban area, with slower social and economic development, hence its urban area was expanding slowly. The largest urban area of Nanchang in 2020 was Nanchang county (208.47 km^2^), followed by Xinjian county (133.34 km^2^) and Qingshanhu (111.87 km^2^). The smallest urban area was Donghu (21.82 km^2^), Xihu (24.79 km^2^) and Qingyunpu (27.50 km^2^). In the last 21 years, the highest rate of cropland loss has been found in Xihu (51.37%), followed by Qingyunpu (49.68%) and Qingshanhu (34.04%) (Figure 5b). Among the nine districts and counties in Nanchang, only Xinjian County has increased its cropland by 4.58%, while the other eight districts and counties all have lost cropland. In addition, the water and grassland areas of Xinjian have decreased by 20.02% and 70.78%, respectively. Urban expansion was the main factor in cropland and ecological land loss. From 2000 to 2020, the forest area of Qingyunpu decreased by 97.33%, followed by Donghu (95.00%) and Xihu (54.72%) (Figure 5c). At the end of 2020, the total forest area of Qingyunpu, Donghu and Xihu was less than 0.01 km^2^, except for Jinxian county and Anyi county, whose forest area increased by 34.04% and 9.29%, respectively; the forest areas of the other seven districts and counties all decreased to different degrees (Figure 5c).

In the last 21 years, cropland, forest and urban land have occupied the largest areas in Nanchang city. Among them, urban land has an average annual growth rate of 3.21%, which was significantly higher than cropland (−0.22%) and forest (0.27%) (Figure 6a,b). Among them, the largest annual change rate of Nanchang was in 2004 (6.48%), followed by 2013 (5.95%) and the smallest annual change rate was in 2020 (0.16%) (Figure 6a). The annual growth rate of urban land has been generally higher than forest and cropland from 2000 to 2020. It has a small fluctuation, and the city showed a stable increasing trend (Figure 6b). The abnormal value is the urban increase rate from 2003 to 2004, during which Nanchang City had the largest growth rate of fixed assets’ investment (67.9%). Thus, the rapid growth of fixed assets investment may be one of the important factors for the rapid urban expansion (Figure 6b). It is worth noting that the distribution of the annual change rate of the forest area has large dispersion and high fluctuation. This may be due to the uncertainty of the land use classification method and classification accuracy. The median line of the annual change rate in cropland is negative and less volatile, indicating that cropland change is dominated by loss and has a stable trend.

From 2000 to 2020, the comprehensive index of land use in Nanchang City has increased from 276.86 to 283.19, indicating a significant increase in land use degree in Nanchang City (Figure 7a). Compared with the non-time series, time series land use shows more details of land use change. For example, the comprehensive index of land use in Nanchang City decreased from 2013 to 2017, and land use was in a period of decline. The comprehensive index of land use increased slowly from 2005 to 2007, indicating that the land use of Nanchang was in the adjustment period. The Pearson Correlation Analysis was employed in this study, and the results show that the correlation coefficients of the comprehensive index of land use and GDP per capita, population and fixed assets investment were 0.92, 0.90 and 0.85, respectively, at 0.01 significance level (Figure 7b). This indicates that the change in the degree of land use is influenced by GDP per capita, population and fixed assets investment. GDP per capita is an important indicator of social affluence, and GDP per capita is an important influencing factor for the increase in land use degree. 

After normalizing the data, this study established a multiple regression model of land use composite index and GDP, and the results showed that the increase in per capita GDP and population was an important factor to promote land intensive use (Table 3). At the same time, the rapid increase in GDP and fixed asset investment will lead to extensive development of land use. In addition, the R squared of the regression model is 0.94.

Cropland was the largest land-use type and urban land was the land use type with the largest change rate in Nanchang. Then, cropland and urban land were employed to discover the regional differences in the changes. The results show that there were significant regional differences between the change in cropland and urban areas in the nine districts and counties of Nanchang (Figure 8). The relative change rate of cropland in Qingyunpu was the largest in 2008 (28.07), and the relative change rate fluctuated widely around the average, followed by Xihu (Figure 8a). The relative change rates of cropland in Qingyunpu and Xihu districts were significantly higher than that in the other seven districts and counties, indicating that the relative change of cropland in Qingyunpu and Xihu districts was drastic, which reflected the drastic loss of cropland and the acuity of land resource development. Moreover, cropland loss in Xihu and Qingyunpu was 51.37% and 49.68, respectively. The urban relative-change rate in Honggutan was largest in 2004 (3.81), followed by 2013 (3.67) and 2015 (3.66), and the urban relative-change rate in Honggutan was much greater than the other eight districts and counties (Figure 8b). From 2000 to 2005, the urban relative-change rate of Qingshanhu has been greater than two, second only to Honggutan and since then the urban relative-change rate of Qingshanhu has been regionally stable and less than 1.5 (Figure 8b). The urban relative-change rate indicated that Honggutan had experienced continuous and high-intensity development of land resources from 2000 to 2016, and its development intensity had been significantly higher than that of other districts and counties. Meanwhile, Qingshanhu has experienced high-intensity development of land resources and urban expansion from 2000 to 2005. The relative change rates of urban land in Anyi County and Xinjian County have increased gradually in the past 20 years, and the intensity of urban expansion has been strengthened. Xinjian County, on the other hand, began experiencing high-intensity development of land resources in 2017, but its development intensity was still lower than that of the earlier development intensity in Honggutan. Therefore, Honggutan was the key area of Nanchang’s urban expansion in the past 20 years. 

### 3.2. Characteristics of Land Use Pattern Change

The Shannon’s Evenness Index (SHEI) at the landscape scale of land use in Nanchang increased by 7.56%, showing a fluctuating increasing trend (Figure 9a). It indicates that the diversity of land use in Nanchang City has increased and the patches’ distribution uniformity of each land use type has also increased. The Patch Density (PD) index has gone through two stages of increasing from 2000 to 2008 and decreasing from 2009 to 2020, and the PD in 2020 is 9.20% higher than that in 2000 (Figure 9b). It indicates that the process of land use change in Nanchang City has gone through two stages of increasing and then decreasing in landscape heterogeneity, which also shows the increasing and decreasing process of human activities. Contagion Index (CONTAG) has decreased by 4.66% from 2000 to 2020, with an average annual growth rate of −0.24%, this means that urban expansion leads to an increase in landscape fragmentation and the deterioration of landscape continuity (Figure 8a and Figure 9c). The number of patches (NP) has increased by 9.20% from 2000 to 2020, indicating an increase in the number of patches of land use in Nanchang City and an increase in landscape fragmentation. In summary, land use in Nanchang City has gone through dramatic changes in the past 21 years. Urbanization and intensified human activities have accelerated the landscape fragmentation of land use spatial distribution and the spatial heterogeneity has significantly increased.

With the social economy developing, urbanization and increased human activities have become the main factors of dramatic land-use changes. Urban land was the only type of land use that has increased substantially among the six land-use types. Therefore, in order to deeply understand the variation of urban land-use types, this study analyzed the change characteristics of urban land in Nanchang. Existing studies are limited by the difficulty of obtaining land use data and untimely updates and are only able to analyze land use at discontinuous times. Unlike existing studies, this study used spatial–temporal Big Data technology of remote sensing to obtain continuous time data of land use rapidly and reliably, and combined it with time series statistics for comprehensive analysis to explore the change characteristics and driving mechanisms of land use. 

The land use with continuous time series of Nanchang City indicated that the urban area has expanded by 87.81% with an average annual growth rate of 3.21% (Figure 10a). The Largest Path Index (LPI) of urban patches increased by 186.49% with an average annual growth rate of 5.48% (Figure 10b), and the largest patch was the central city of Nanchang, indicating a significant trend of urban expansion in the central city. It is noteworthy that the largest LPI growth rate was in 2002 and 2008 at 15.49% and 15.89%, respectively, which implied significant urban expansion. The fixed assets’ investment in Nanchang City increased significantly in 2002 and 2008, 41.40% and 34.00% higher than in 2003 and 2009, respectively, and continued to grow at a high rate for four and three years from then on. Thus, we have reason to believe that the increase in fixed assets investment contributed to the rapid increase in LPI of the urban patches and urban expansion (Table 4). Existing studies also show that economic development is directly related to the urban landscape pattern [36,37]. The regression coefficient of LPI with GDP, GDP per capita and fixed investment was calculated, and the total R square is 0.99 (Table 4). This shows that POP and fixed investments are the key factors influencing the change of LPI.

The Patch Cohesion Index (COHESION) reflects the degree of connection between urban patches, and the larger the value, the stronger the spatial connection between patches. The COHESION of Nanchang City has increased by 3.30% from 2000 to 2020, with an average annual growth rate of 0.16%. Among them, the average annual growth rate from 2000 to 2008 was 0.33% and from 2009 to 2020 was 0.05% (Figure 10c). This indicates that urban areas have expanded rapidly from 2000 to 2008, and there was a strong spatial connection of urban expansion. This implies that the direction of urban expansion is closely related to spatial factors such as the distance of existing urban land space. The spatial relationship between urban expansion and the existing urban area has weakened in 2009–2020, implying that urban expansion was mainly punctiform and there was no strong spatial linkage between patches. Moreover, the Splitting Index (SPLIT) is one of the important indicators to measure the degree of landscape separation, and the larger the value, the higher the separation degree between landscape patches and the more fragmented the spatial distribution. The SPLIT of urban land-use types in Nanchang City has decreased by 90.48% from 2000 to 2020, indicating that the separation degree of urban land in Nanchang City has decreased and the fragmentation degree of urban spatial distribution has also decreased, which means that the aggregation degree of urban spatial distribution has increased (Figure 10d).

There were four hot spots of land use change (HSLUC) in Nanchang, which were located in the northeast, northwest, central and southeast of Nanchang (Figure 11). Only the HSLUC in northeast China was formed because the project of converting cropland to lakes transformed cropland into wetland. Urban expansion is the main reason for the formation of the other three HSLUC. Among them, the central urban area of Nanchang city was the main HSLUC, its urban area expansion was the largest, and the land use change was the most drastic.

### 3.3. Uncertainty

This study used remote sensing spatial–temporal Big Data and machine learning methods to obtain continuous time-series data of land use and analyzed the spatial–temporal pattern change of land use in Nanchang City with time series statistical data. We bridged the gap of time series land-use research based on a statistical perspective. However, there are still uncertainties in the results of this study, mainly concerning the uncertainty of the method and the uncertainty of the data accuracy.

One of the main uncertainties of the method is the limitations in the selection of training datasets, which cannot extract all features and use them for model training. As a result, there is uncertainty in the accuracy of the model. Furthermore, the land use data have their uncertainties of accuracy as the third party of sample data, and we cannot guarantee that the precision of the sample data is completely accurate. Data errors may be passed on to the classification data of land use in this study, leading to uncertainty in the results of this study. In the subsequent study, we can improve the classification method of land use and select land use data with higher accuracy and credibility as the training sample data to reduce the uncertainty of the study results.

## 4. Conclusions

Land use in Nanchang City has changed dramatically from 2000 to 2020. Among them, the urban area has increased enormously, and the area of cropland, on the other hand, has decreased significantly. Meanwhile, most of the reduced cropland area has been transformed into urban land. This study provides more evidence for the loss of cropland caused by urban expansion. In addition, there were significant differences in land use changes among the nine administrative units in Nanchang. Among them, Honggutan showed dominance in urban expansion, while the old city showed a significant loss of forest areas which were converted into urban use. Over the past 21 years, land use changes in Nanchang have been dominated by urban expansion, which led to the loss of cropland and forest. The spatial pattern of land use has changed significantly, and the urban spatial pattern has become more aggregated, while the spatial distribution of cropland, forest and grassland has become more fragmented. The landscape pattern of land use has been more fragmented, and the city has expanded rapidly. Moreover, there was a significant correlation between social–economic development and land use level, and the social economy was obviously correlated with the landscape patterns of land use. Population, GDP and fixed assets investment were the three main drivers of land use change. They affected the area change of land use types, as well as the form and pattern of land use spatial patterns, and further affected the development of the city. Therefore, understanding the characteristics of land use change can help to clarify the driving mechanisms of land use change and provide support for government decisions and the rational and efficient use of land resources.

## Figures and Tables

**Figure 1 ijerph-20-02330-f001:**
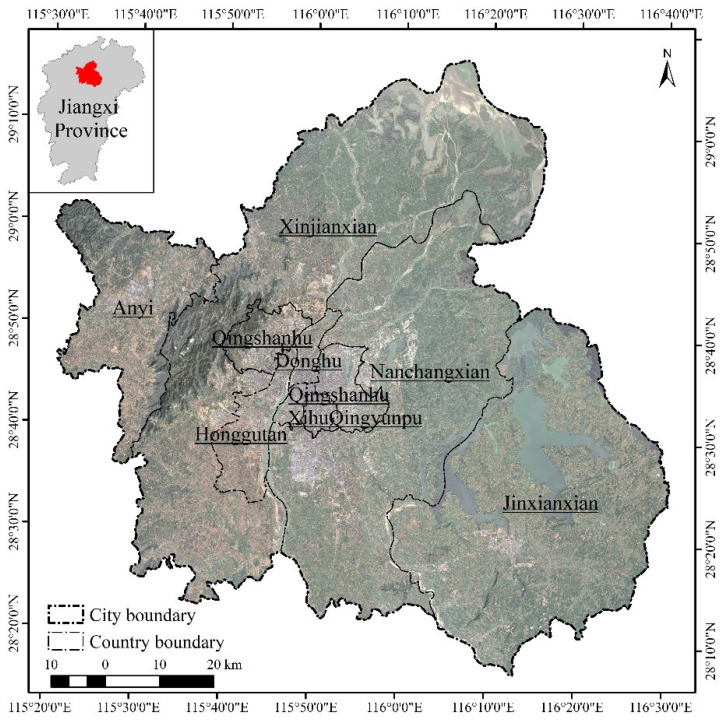
Nanchang City.

**Figure 2 ijerph-20-02330-f002:**
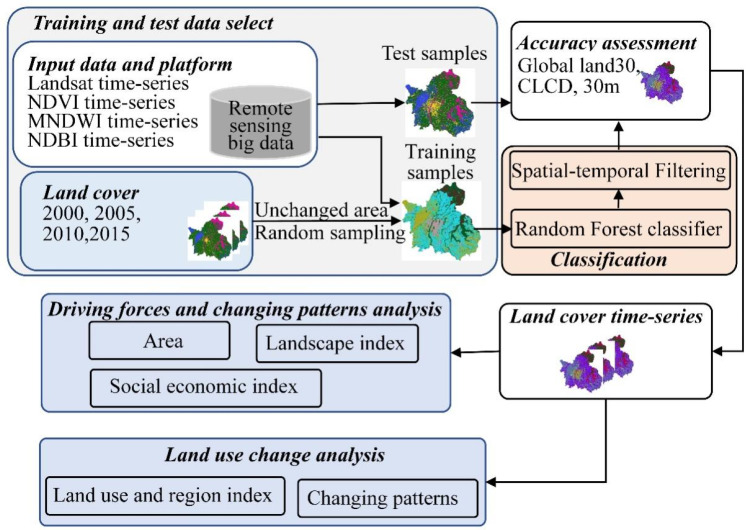
Research framework diagram.

**Figure 3 ijerph-20-02330-f003:**
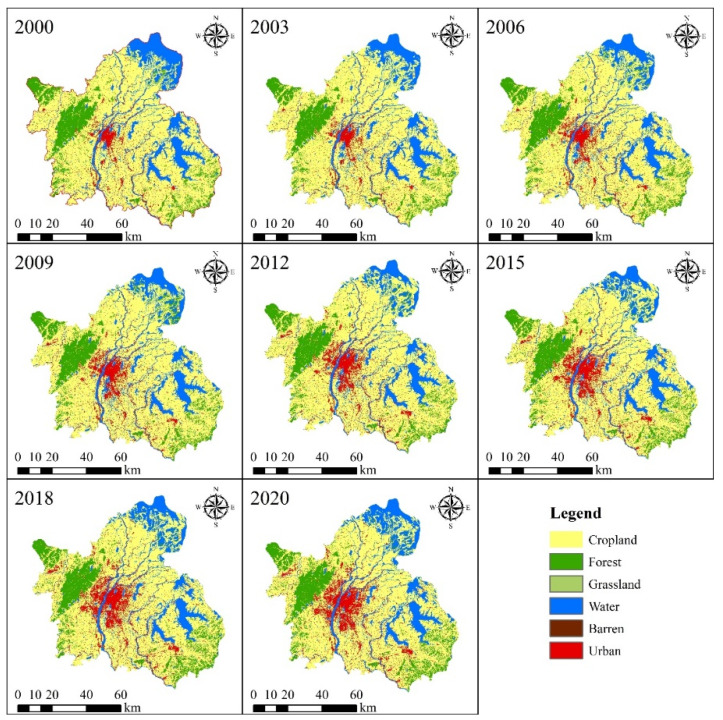
Nanchang City Land Use Map 2000–2020.

**Figure 4 ijerph-20-02330-f004:**
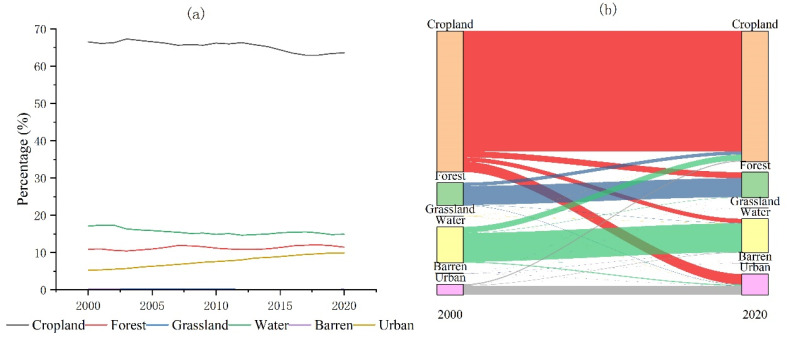
Land use statistics and change map for Nanchang City 2000–2020. (**a**) Summary of the area of six types of land use; (**b**) The 2000–2020 land use transfer map.

**Figure 5 ijerph-20-02330-f005:**
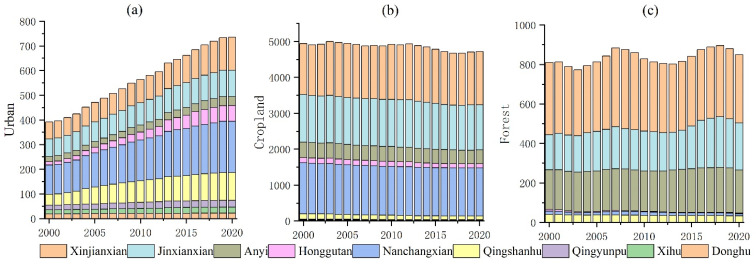
Area of urban, cropland, forest changing patterns (unit: km^2^). (**a**) changing patterns of urban area from nine counties; (**b**) changing patterns of cropland area from nine counties; (**c**) changing patterns of forest area from nine counties.

**Figure 6 ijerph-20-02330-f006:**
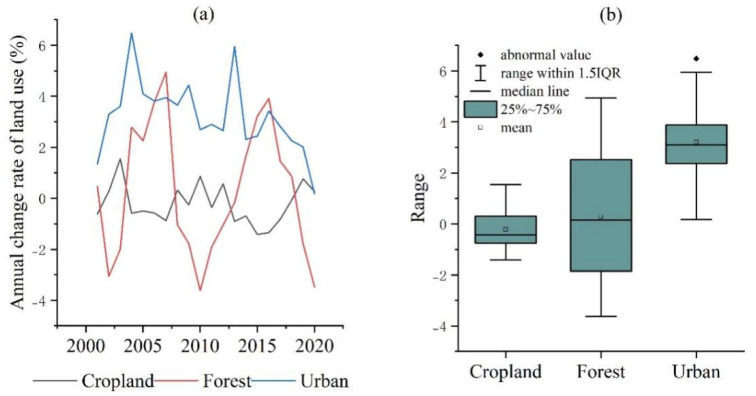
Cropland, forest, urban annual rate of change of land use (%). (**a**) Cropland, forest, urban annual change rate (%); (**b**) Box plot of the annual change rate.

**Figure 7 ijerph-20-02330-f007:**
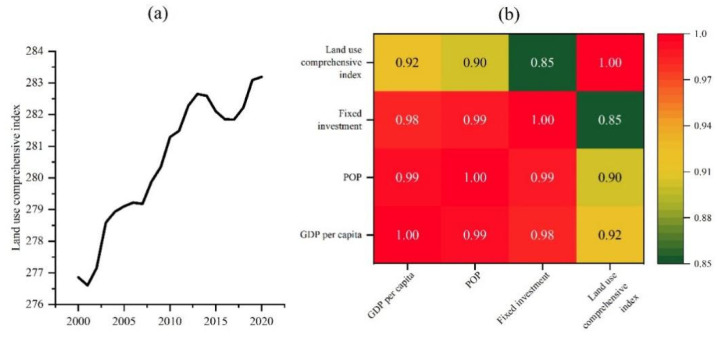
Comprehensive index of land use and its social economic-correlation. (**a**) Comprehensive index of land use from 2000 to 2020; (**b**) Correlation coefficient between the comprehensive index of land use and social-economic indicators, all the *p* < 0.01.

**Figure 8 ijerph-20-02330-f008:**
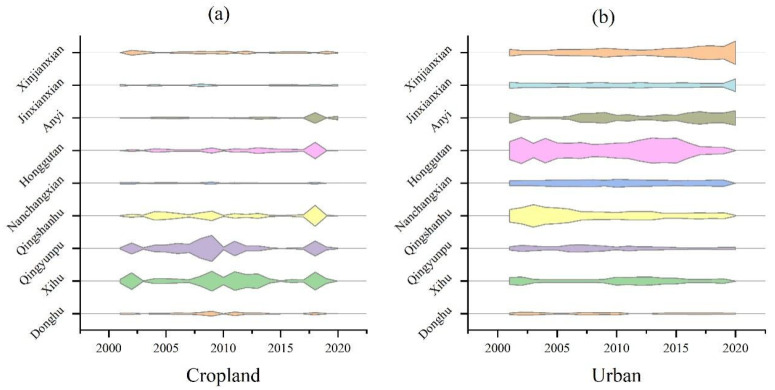
Kite diagram of relative change rates in cropland and urban land use areas. (**a**) the relative change rates in cropland; (**b**) relative change rates in urban.

**Figure 9 ijerph-20-02330-f009:**
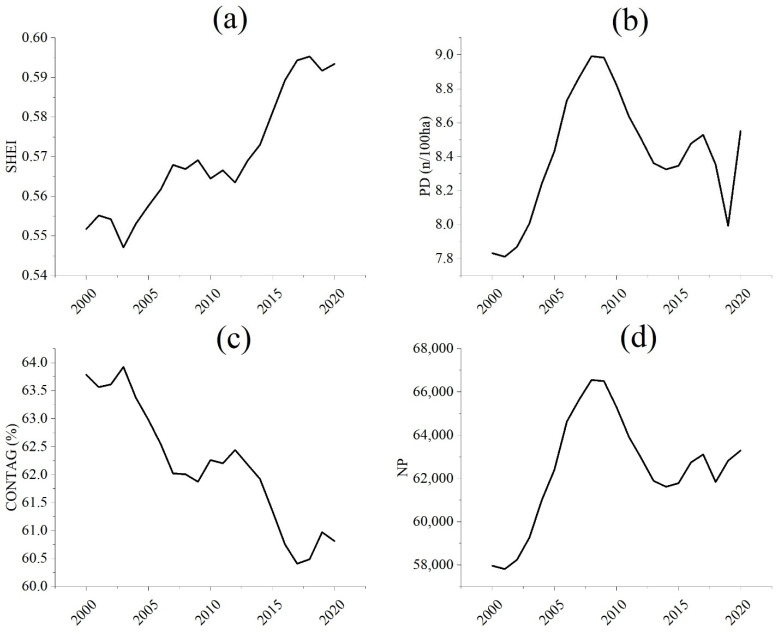
Landscape index change of land use in Nanchang, 2000–2020. (**a**) the Shannon’s Evenness Index (SHEI) of Nanchang from 2000 to 2020; (**b**) the Patch Density (PD) index of Nanchang from 2000 to 2020; (**c**) the Contagion Index (CONTAG) of Nanchang from 2000 to 2020; (**d**) the number of patches (NP) of Nanchang from 2000 to 2020.

**Figure 10 ijerph-20-02330-f010:**
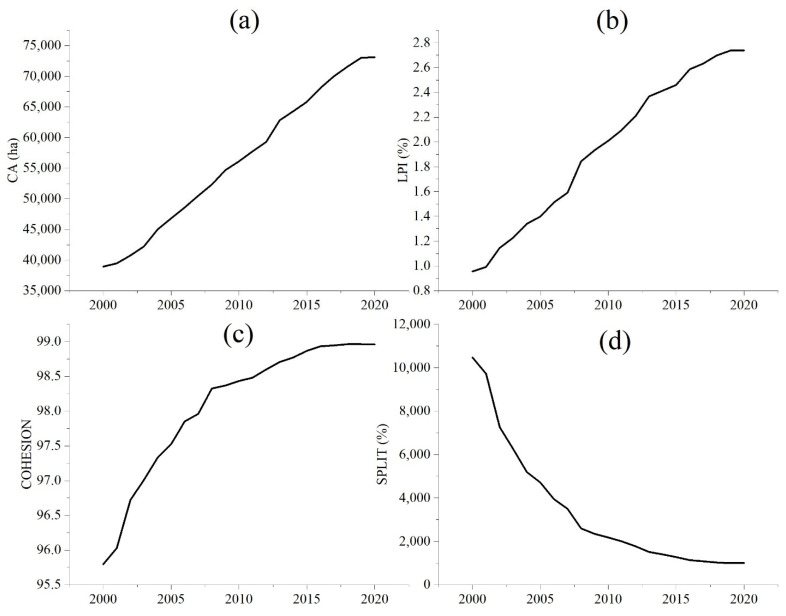
Changes in urban landscape pattern index. (**a**) urban landscape area; (**b**) the Largest Path Index (LPI) of urban patches; (**c**) the Patch Cohesion Index (COHESION) of urban; (**d**) Splitting Index (SPLIT) of urban.

**Figure 11 ijerph-20-02330-f011:**
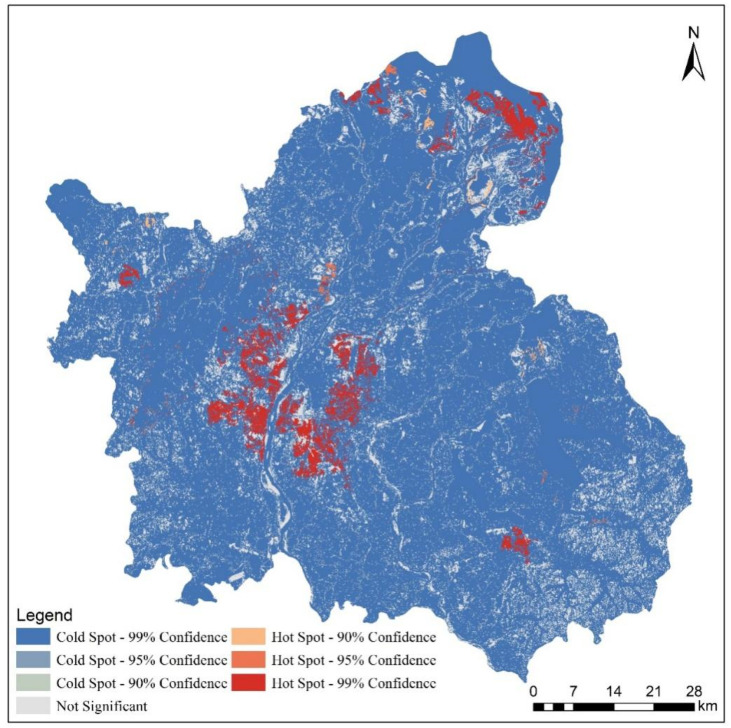
Hot spot of land use change from 2000 to 2020.

**Table 1 ijerph-20-02330-t001:** The classification values of land use degree.

Type of Land Use	Barren	Forest, Grassland, Water	Cropland	Urban
Classification index	1	2	3	4

**Table 2 ijerph-20-02330-t002:** Land use transfer matrix (unit: square kilometers).

	2020	Cropland	Forest	Grassland	Water	Barren	Urban
2000							
**Cropland**		4320.89	122.25	1.96	266.18	0.96	9.90
**Forest**		178.63	654.66	0.13	8.87	0.00	0.37
**Grassland**		1.10	1.27	0.65	0.80	0.91	0.07
**Water**		118.48	8.10	3.50	952.10	4.93	11.26
**Barren**		2.02	4.63	0.33	1.39	2.06	0.09
**Urban**		318.62	11.33	2.81	29.54	2.90	357.46

**Table 3 ijerph-20-02330-t003:** Regression coefficient.

	Regression Coefficient	Standard Error	Significance
GDP	−2.06	2.06	0.00
GDP per capita	3.26	1.19	0.00
POP	0.39	0.54	0.00
Fixed investment	−0.77	0.98	0.00
Intercept	0.08	0.05	0.01

**Table 4 ijerph-20-02330-t004:** Regression coefficient of LPI with GDP, GDP per capita and fixed investment (R square is 0.99).

	Regression Coefficient	Standard Error	Significance
Intercept	0.01	0.02	0.43
GDP	−3.79	0.67	0.00
GDP per capita	3.50	0.39	0.00
POP	0.66	0.18	0.00
Fixed investment	0.61	0.32	0.01

## Data Availability

The data are from the Nanchang Statistical Yearbook, the China Geographic Information Bureau, the Resource and Environment Science and Data Center, and Google Earth Engine platform.

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
