# Peer review of "The Spatio-Temporal Patterns and Driving Forces of Land Use in the Context of Urbanization in China: Evidence from Nanchang City"

_ijerph, 2023, doi:10.3390/ijerph20032330_

Round 1

Reviewer 1 Report

Int. J. Environ. Res. Public Health

Manuscript Number: IJERPH-2093924

The spatio-temporal patterns and driving forces of land use in the context of urbanization in China: An evidence from Nanchang city

Review comments

The manuscript addresses land use changes and drivers over the last two decades in the city of Nanchang in China. The study contributes with valuable information in terms of the approach and findings. The research framework and methodologies adequate to assess the spatio-temporal patterns and driving forces of land use. My major concerns are related to the structure and organisation of the research and, following that, the unbalance of the sections presenting the study:

(a)Introductionis too long and needs to be re-written. Authors should contextualize the study, the aims and the purpose of the research in a more concise way.

(b)Results and discussionis actually a results section. Again there is information that can be associated to the legend of the figures and, with it, the text be focused on the main information to document the results, which will improve the readability.

(c)Discussioncorresponds to an important section of the study that also needs to be re-written. Authors should confront the results with the information available (scientific and available reports). Eventually some information considered in the introduction can be useful to confront with the results obtained. I also recommend Authors to consider the limitations of the study.

(d) Finally, the manuscript needs a deep proofreading in English; there are many typos along the text. There is no reason for the abstract be presented in present the results are presented in past sentence…

Author Response

The responses please see attachment.

Reviewer 2 Report

Review comments for IJERP 2093924

P1.  Line 2, comma needed before “human activities”.  First paragraph is too long. Has a natural break at the word “Since” in line 1 of page 2. Start a new paragraph there.

P1. “Traditional land use research is prone to implement land use surveys,” What I think you mean is: Traditionally, land use research has emphasised land use surveys… I am not sure that it is appropriate to refer to utilization planning as land use research. Is this exactly what Tarawally el at said?

P2. Para 1. Population of the world is about 8 billion – it is estimated to have passed 8 billion just a few weeks ago. The statement here implies it is about 4 billion. About 4.4 billion people (56%) live in urban areas today. https://www.worldbank.org/en/topic/urbandevelopment/overview#:~:text=Today%2C%20some%2056%25%20of%20the,people%20will%20live%20in%20cities.

P.2. , “loss of cropland (Qiu et al., 2020), etc”. Rephrase as:…and loss of cropland (ref). You have already said “such as”, so the etc is unnecessary (and is inappropriate in a scientific report).

P2. “Liu et al (Liu et al., 2005) explored the relationship between urban expansion and population change, monetary growth and land use policy changes in 13 major cities in China, and the results showed that urban expansion is influenced by a combination of population, monetary and land policies.”  Rewrite as:

 Liu et al (2005) showed for 13 cities in China that urban expansion is influenced by a combination of population, monetary and land policies.

The second half of this last paragraph on p.2 actually says very little of interest. Please summarise and shorten to make a key point, or leave it out. I have to disagree with you on time series data versus static data. All spatial data showing change are presented as time series, even if the time period of data collection is short.  Your own presentation may cover a longer time series, but it still uses a series of snapshots (e.g. on a year-by-year basis; your figure 3 shows change on a 3-year basis), so is not strictly “continuous”. The implied criticism of earlier studies is not interesting, and is arguably wrong. The implication that your study is somehow better because it covers a longer time period (which you argue in several places) is both arrogant and inappropriate. Please take these comments out of the paper. There is no question that data sets are becoming both larger (in terms of quality and density of information) and longer (in terms of time series). But it is inappropriate to argue that this development makes the analyses and interpretations somehow better.

p.3, second paragraph. Again, you are arguing that data sets are getting better, without actually giving the reader any useful information. There is no argument about this, and there is no need to spend an entire long paragraph saying so. Take this paragraph out, or summarise it to tell the reader what others have found. Please stop just telling the reader what others are doing and how wonderful it is.

P.3. Last line. RMB is not a well-known reference for the Chinese currency internationally. I had to look it up. Perhaps it would be more appropriate to use the symbol, which most people would recognise: ¥

Section 2.3.3, first paragraph. After reading this important but very confusing description several times, I believe that the primary confusion lies in the reference to the study area being a hot spot (presumably in relation to elsewhere in the country) combined with multiple references to ”the study area”. Scale is important here, and the scale of “local” relative to the scale of the overall study area is not defined. How was “local” defined, and how many calculations enter the index calculation? Lastly, if the aim is to reveal regional differences, as stated in the last sentence, then presumably that is to be done elsewhere or by someone else. My interpretation is that this is one region under study? But perhaps I have misunderstood the relationship between “local” and “study area”.

p.7. 66.49%-63.57% is not 4.39%

p.7. Main paragraph. This paragraph summarises the data in Figure 4, but I am afraid that it is very confusing. After reading it several times, I believe that Fig 4b should be removed and replaced by a Table that puts the %s in the text into a format that can be understood. Many %s are quoted, but they are not all calculated from the same base, although that is not explained. Fig 4b simply cannot be understood without considerable explanation, and the text as presented does not help. Fig 4a is useful, although its scale is dominated by cropland, which makes it hard to see the trends. I wonder if cropland should be removed and then perhaps the other categories could be included.  Also, the lines should be given patterns – the colours are not easily distinguished, and anybody printing this paper in black and white will not be able to see the trends at all.

p.8. How do the nine administrative districts differ from “local”, or are they the same?

p.8. Another long description of %s that would be much easier to understand if converted into a table.

Figure 5. Presumably the data in these graphs are calculated on an annual (yearly) basis? Is the variation here of interest, as your written description focuses on the beginning and end of these lines? Ah, I see the variation is explored in Figure 6.  Fig 6b adds no extra information and should be removed.

p.10. Top. Here you give several sentences explaining why your analysis is better than anybody else’s. Please take this out.

p.10. While it makes sense that GDP, population and fixed asset investment are correlated, it does not follow that GDP per capita should increase as the other two increase. This is a somewhat curious result, although it suggests that standards of living are increasing along with population and investment. Perhaps that is linked to increasing urbanism?

p.10: “One of the innovations of this study is to quantify the correlation between land use change and social economy using continuous time series land use data and statistical methods. Therefore, the correlation among comprehensive land use index, per capita GDP, population and fixed asset investment was analyzed in this study. The Pearson Correlation Analysis was employed in this study.”

The above is all repetition of Methods. It is not results. It is also claiming (again) that this study is better than others. Please take it all out.

p.10: “This indicates that the change in the degree of land use is influenced by GDP per capita, population and fixed assets investment. GDP per capita is an important indicator of social affluence, and GDP per capita is an important influencing factor for the increase in land use degree. In summary, GDP per capita, population, and fixed assets investment are three important drivers of land use change in Nanchang City.

You have said the same thing over and over again here. Please remove the repetition. Also, unless you have some reasons for needing to use a non-parametric analysis, this analysis should probably be a multiple regression rather than three separate correlation coefficients across the three inter-related variables. It is not appropriate to conclude that one of the factors is dominating the influence here (as you suggest with GDP per capita). These three factors are correlated, which is not very surprising given the many likely connections among them.

p.10:  “A large number of existing studies of land use cannot quantify the correlation between land use change and social economy using statistical methods due to the lack of continuous time series data on land use,”

This statement is discussion. It does not belong in the Results. Also, a continuous time series is an arbitrary concept. What matters for a correlation analysis is that there be a reasonable number of points along an axis (such as time). The time interval does not matter (it could be every ten years over 100, or every year over ten; yours is 20 years, and the time intervals of your analyses vary). These arguments are particularly not appropriate in the Results section.

Fig 7 needs considerable work on some of the detail (overlapping labels, “capital” [capita]. But it is not appropriate to present a whole series of separately derived correlation coefficients when there are better techniques that avoid the problems of multiple univariate statistical tests. Use multiple regression or something similar. Please remove the table of coefficients.

Top p. 11. Please focus on the results. When calculated on a % basis, cropland appears to have a small decrease while urban appears to have a large increase. The reality is that the absolute areas involved may be quite similar. Urban is growing – no doubt due to population increase, increased local investment, and perhaps government policy. That land has to come from somewhere and the most likely source is cropland. What is interesting here is the effects of this change on land use, and the associated cultural, social and economic implications. I am not sure that there is anything new in this last section of the results that has not already been demonstrated? This section reads like discussion and should probably be taken out unless there is something genuinely new that I have missed.

p.12. “Unlike existing studies, this study uses spatial-temporal big data technology of re[1]mote sensing to obtain continuous time data of land use rapidly and reliably, and com[1]bines it with time series statistics for comprehensive analysis to explore the change char[1]acteristics and driving mechanisms of land use.”

This is a repeat of Methods, and is a claim that this study is better than others. Please take it out.

p.12. “It is noteworthy that the largest LPI growth rate is in 2002 and 2008 at 15.49% and 15.89% respectively, which implies significant urban expansion. The fixed assets investment in Nanchang City increased significantly in 2002 and 2008, by 41.40 % and 34.00 % respectively over the previous year, and continued to grow at a high rate for four and three years respectively. Thus, we have reason to believe that the increase in fixed assets investment has contributed to the rapid increase in LPI of the urban patch Int. J. Environ. Res. Public Health 2020, 17, x FOR PEER REVIEW 13 of 17 13 and urban expansion.”

I wonder if the increases in fixed asset investment is directly connected to the large increases in urban growth rate. In other words, the building and other infrastructure investment that accompanies urban expansion is how the higher levels of urban growth were achieved. If so, it is hardly appropriate to conclude that they are connected. Clearly, they are directly connected. As also, are they directly connected with GDP. Investment is required to create urban growth, land will therefore be taken to support that investment, and the increased commercial activity creates higher GDP. There are some grammatical errors in the rest of this very long paragraph, and some of it is very difficult to follow. It needs considerable work to communicate effectively. A start would be to break it up into several paragraphs.

He last paragraph about hot spots seems to be repetitive and unnecessary. What is the result here?

Discussion

Take out section 4.1

It bothers me that the main conclusion in the discussion is the very uninteresting statement that good planning is needed, including taking into account long-term trends. There is nothing new in this conclusion, which is the same as would be stated in any Planning 101 course. There is no useful policy direction here. As far as I can tell, the government of China does better planning than most countries, due to top-down political structures and decision processes. The most difficult part of planning is predicting long-term requirements, and in more democratic societies, the 3-4 year electoral cycle tends to force short-term decision-making on policy people. China does not have that problem. However, there is no question that current trends in population patterns may not be sustained (particularly a large proportion of people heading into old age). World population must stabilise (and even reduce) if we are to live sustainably on the planet, so some city loss and/or shrinking is inevitable. If the authors want to explore this issue, it would be interesting to do so from the Chinese perspective, but unfortunately this discussion barely touches on the relevant issues. This paper is about growth. The implication that this growth might not be sustainable is interesting and worth exploring as a discussion topic.

Author Response

The responses please see attachment.

Reviewer 3 Report

Figure 4 should be improved. The information given by the picture is not clear. A better option would be to depict the area on side (a) in relative terms of increase (e.g. from 100% up to... or down to....). Also, part (b) is confusing.

The formulation on page 9/17 "In the last 21 years, Cropland, Forest, and Urban are the three largest land use types in Nanchang city with an average annual growth rate of 3.21%, which is significantly higher than Cropland (-0.22%) and Forest (0.27%) (Fig.6 a-b)." is not clear and it should be revised.

Figure 5 should be improved. The information given by the picture is not clear. A better option would be to depict the area on side (a) in relative terms of increase (e.g. from 100% up to... or down to....).

Figure 6 (the star graph) is difficult to read. A better fit would be a simple XY graph. 

Figure 7 since ** p< 0.01 comment is valid for all situations, it make little to no sense to put it in all places. It can be mentioned explicitly in figure title that it is valid for all cases

Author Response

The responses please see attachment.

Round 2

Reviewer 1 Report

Authors did improve the manuscript. However, there are two references to ‘Discussion’. I recommend Authors to analyse the data from a critical perspective to science. If there is not so much information available on the topic for the regions to compare then better to merge Results and discussion, which can also include subtopics.The introduction is still too long; is should be more concise and some of the information eventually proper for the discussion.

Author Response

Comment: Authors did improve the manuscript. However, there are two references to ‘Discussion’. I recommend Authors to analyse the data from a critical perspective to science. If there is not so much information available on the topic for the regions to compare then better to merge Results and discussion, which can also include subtopics.The introduction is still too long; is should be more concise and some of the information eventually proper for the discussion.

Response to comment: Thank you for your time and suggestion. We reorganized the results and discussion sections and removed superfluous topics. In addition, we shortened and rewrote the introduction. (Page 1-4)

Reviewer 2 Report

p.2. "Researchers have been studying urban expansion since the 1820s (Liu et al., 2021), and they mainly focused on urban morphology and spatial structure in land use and proposed various theories, hypotheses and models of urban expansion, such as the concentric model, sector theory and multi-core model (Dai, 2020)"

Not a sentence. suggest rewrite: ...focussing mainly on urban morphology and spatial structure in land use. Suggested theories...expansion include the concentric...

p.7. The reader gets very confused with the various figures (using mostly percents) and land area statements in this section of results. An example is  32.17% of grassland is converted to urban, and 10.40% of 1.22 km2 of barren is converted to grassland (=0.1 km2 ?). In the context of the land areas quoted here, this seems very small and possibly uninteresting. The two decimal places being quoted are not appropriate for such small areas of land.  Table 2 is especially confusing because no units are given. Are these square km, or percents, or perhaps even some kind of calculated index. Better consistency in quoted figures is needed here. 

p.10 (near top). "Land use" cannot decline. Rate of change of land use can decline. 

I am afraid that I still find the discussion uninteresting, with the statements very obvious and lacking useful insight. As your results section is both results and discussion, I think this final section of discussion should be left out. 

Author Response

Comment 1: p.2. "Researchers have been studying urban expansion since the 1820s (Liu et al., 2021), and they mainly focused on urban morphology and spatial structure in land use and proposed various theories, hypotheses and models of urban expansion, such as the concentric model, sector theory and multi-core model (Dai, 2020)"

Not a sentence. suggest rewrite: ...focussing mainly on urban morphology and spatial structure in land use. Suggested theories...expansion include the concentric...

Response to comment 1: Thank you for your time and suggestion. We rewrote this sentence according to your suggestion. (Page 2 Line 36-39)

Comment 2: p.7. The reader gets very confused with the various figures (using mostly percents) and land area statements in this section of results. An example is  32.17% of grassland is converted to urban, and 10.40% of 1.22 km2 of barren is converted to grassland (=0.1 km2 ?). In the context of the land areas quoted here, this seems very small and possibly uninteresting. The two decimal places being quoted are not appropriate for such small areas of land.  Table 2 is especially confusing because no units are given. Are these square km, or percents, or perhaps even some kind of calculated index. Better consistency in quoted figures is needed here.

Response to comment 2: Thank you for your time and comments. I'm sorry that our inaccurate description has confused you. As a response, we rewrote these sentences, and replaced figures with land area. In addition, we increased percentages after some figures in order to allow the readers to understand the meaning of the sentence accurately. (Page 8 Line 16-21)

Comment 3: p.10 (near top). "Land use" cannot decline. Rate of change of land use can decline.

I am afraid that I still find the discussion uninteresting, with the statements very obvious and lacking useful insight. As your results section is both results and discussion, I think this final section of discussion should be left out.

Response to comment 3: Thank you for your valuable suggestion. We rewrote the sentence based on your suggestion. Moreover, we deleted the final section of discussion, and reorganized the Part 3 “Results and discussion”. (Page 10 Line 23-25; Page 16)
